# Phase-Matching in Nonlinear Crystal-Based Monochromatic Terahertz-Wave Generation

Pengxiang Liu [1,2], Chuncao Niu [1,2], Feng Qi [2,3,4,*], Wei Li [1,2], Weifan Li [2,3,4], Qiaoqiao Fu [2,3,4], Liyuan Guo [2,3,4] and Zhongyang Li [5]

1 College of Information Engineering, Shenyang University of Chemical Technology, Shenyang 110142, China
2 Shenyang Institute of Automation, Chinese Academy of Sciences, Shenyang 110169, China
3 Key Laboratory of Liaoning Province in Terahertz Imaging and Sensing, Shenyang 110169, China
4 Key Laboratory of Opto-Electronic Information Processing, Chinese Academy of Sciences, Shenyang 110169, China
5 College of Electric Power, North China University of Water Resources and Electric Power, Zhengzhou 450045, China
* Correspondence: qifeng@sia.cn

**Abstract:** Optically pumped nonlinear frequency down conversion is a proven approach for monochromatic terahertz (THz)-wave generation that provides superior properties such as continuous and wide tunability as well as laser-like linewidth and beam quality. Phase-matching (PM) is an important connection between the pump sources and nonlinear crystals and determines the direction of energy flow (as well as the output power). In past decades, a variety of peculiar PM configurations in the THz region have been invented and are different from the traditional ones in the optical region. We summarize the configurations that have been applied in nonlinear THz-wave generation, which mainly fall in two categories: scalar (collinear) PM and vector PM (including macroscopic noncollinear PM and microscopic vector PM). The development of this technique could relax the matching conditions in a wide range of nonlinear crystals and pump wavelengths and could finally promote the improvement of coherent THz sources.

**Keywords:** coherent terahertz source; nonlinear optics; phase-matching; nonlinear optical crystal; optical frequency conversion





## 1. Introduction

Terahertz (THz) technology [1] has attracted considerable interest owing to the unique properties of this band of electromagnetic waves: (1) resonance with vibrational and rotational modes of typical large molecules, (2) spectral overlap with the background radiation of the space, (3) penetration through general non-polar materials (e.g., packages), and (4) low photon energy (unionized to biological tissues). A variety of applications in analytical chemistry [2], astronomy [3], the quality control of industrial products [4], biomedical imaging and diagnose [5], etc., have demonstrated the effectiveness of this technology. As active spectral analysis systems, Fourier transform infrared (FTIR) and time-domain spectroscopy (TDS) are well-established and often utilized. Those methods based on widely tunable and monochromatic THz sources (i.e., frequency-domain spectroscopy) are alternative tools to FTIR and TDS due to their high spectral resolution.

Nonlinear optical frequency down conversion from the laser to the THz wavelength, commonly referred to as difference frequency generation (DFG) or stimulated polariton scattering (SPS), is one of the most effective approaches and could provide the following merits: the availability of a compact pump laser and nonlinear crystals, operation at room temperature, continuous and wide tunability, and laser-like linewidth and beam quality. As discussed by C.M. Armstrong [6], "for some applications, the source's spectral purity, tunability, or bandwidth is more important". These kinds of coherent sources have been

successfully applied in various fields for applications such as the identification of gas species [7], quality evaluations of pharmaceuticals [8], and nondestructive inspection via spectroscopic imaging [9].

In the past few decades, a number of efforts have been made to improve the performance of optical THz sources: from pump lasers via nonlinear crystals to enhancement configurations (including surface-emitted [10], external cavity [11], and waveguide [12] configurations, among others). In the down conversion process, the lasers and THz-waves are mainly coupled through nonlinear susceptibility $\chi^{(2)}$, so phase-matching (PM) is a key factor. It determines the output characteristics of the THz source according to these aspects: (1) the direction of energy flow during the three-wave interaction, (2) the available effective nonlinear coefficient $d_{\mathrm{eff}}$, (3) the suitable pump wavelength (if the laser is powerful), (4) the matchable tuning range, and (5) the tolerance of the angle, wavelength, or temperature (if precise control is necessary). Numerous PM geometries have been adopted according to the optical properties of nonlinear crystals. Some of them (e.g., birefringent, noncollinear, and quasi-PM) are extended from traditional optical regions; some, were original developments (such Cherenkov-type and front-tilting PM [13,14]).

We classify these geometries into two main categories: scalar and vector PMs. In the case of scalar PM, the three beams are collinear (forward or backward output coupling [15,16]). In the other category, a wavevector triangle is properly designed, and the THz output is commonly coupled in a lateral direction. The vector PMs are further divided into two types: macroscopic noncollinear PM (quasi-plane waves in obviously different directions [17]) and microscopic vector PM (between spatial Fourier components [18,19]).

This paper is organized as follows. First, we introduce the basic problem of PM in THz-wave generation and compare it to that in the optical region in Section 2. A brief review of PM geometries obtained via various methods is presented in Sections 3 and 4. The mechanisms of each method is described, and the suitable nonlinear crystals and typical reports are presented. Here, we mainly discuss the interaction between two monochromatic optical waves rather than the optical rectification of ultra-short pulses. Section 3 concerns the scalar (collinear) forms, including birefringent PM in traditional infrared crystals (GaSe/ZnGeP$_2$), the cross-reststrahlen band PM in zinc blende or organic crystals, and quasi-PM (QPM) in ferroelectrics or zinc blende crystals. The vector forms are considered in Section 4, which consists of two Sections 4.1 and 4.2, for the macroscopic noncollinear and microscopic vector PMs, respectively. In Section 4.2, Cherenkov-type, surface-emitted QPM and modal PM are considered. Moreover, other configurations are also enumerated. A summary of the current progress and perspectives is given in Section 5.

## 2. Basic Problem of Phase-Matching in THz-Wave Generation

The down conversion process is, in essence, energy flow from a high-frequency optical wave (called a "pump") to a low-frequency optical wave (called a "seed" or "Stokes") and a THz-wave via non-resonant $\chi^{(2)}$ coupling [20]. The delivered THz frequency depends on the seed wavelength or cavity feedback, which obeys the energy conservation relationship. The configurations of dual-wavelength input are referred to as DFG or injection-seeded THz parametric generation (is-TPG). If the intensities of the two beams are comparable, it is called DFG, and if one is much stronger than the other, it is called is-TPG. In the case of single-wavelength input, a cavity is usually necessary. This process is initially called THz parametric oscillation (TPO) and has recently been called SPS. Although TPO/SPS is considered to be a mixing of second-order parametric processes and third-order Raman scattering in some literature [17], energy conservation and PM still dominate.

The concept of PM has been explained from three points of view: (1) a constructive interference between the fields driven by polarization at different propagation steps [21], (2) momentum (wave vector) conservation, and (3) phase-dependent energy flow from polarization to the field based on Poynting's theorem [22]. Mathematically, the PM condition is expressed as

$$\Delta \vec{k} = \vec{k}_{\mathrm{P}} - \vec{k}_{\mathrm{s}} - \vec{k}_{\mathrm{T}} = \vec{0} \tag{1}$$

where the subscripts p, s, and T denote the pump, seed/Stokes, and THz-waves, with $\omega_p = \omega_s + \omega_T$ and $\omega_p > \omega_s \gg \omega_T$. In a scalar (forward collinear) form, (1) is equivalent to

$$n_p\omega_p - n_s\omega_s - n_T\omega_T = (n_p - n_s)\omega_s + (n_p - n_T)\omega_T = 0 \tag{2}$$

Then, the problem of PM turns into a problem of refractive index matching, which is greatly influenced by the dispersion property of nonlinear crystals.

For a comparison with the traditional optical region (all three waves in one normal dispersion region), we introduce the subscripts 3, 2, and 1 to denote three optical waves, with $\omega_3 = \omega_2 + \omega_1$, $\omega_3 > \omega_2 \geq \omega_1$, and $n_3 > n_2 \geq n_1$. As a result, the collinear wave vector mismatch $\Delta k_{opt}$ is always positive, and a vector triangle will not help to make it zero if all of the refractive indices are on one dispersion curve (isotropic medium or the same polarization).

The wavevector mismatch can be expanded as another form:

$$\Delta k = k_p - k_s - k_T = \frac{\partial k}{\partial \omega}(\omega_p - \omega_s) - \frac{n_T}{c}\omega_T = \frac{\omega_T}{c}(n_g - n_T) \tag{3}$$

where $n_g = c(\partial k/\partial \omega)$ is the optical group index.

The PM in the THz region differs from that in the optical region in two main ways. First, the THz and pump frequencies lie across one or more absorption bands (the reststrahlen band), which breaks the strict relationship above: $\Delta k_{opt} > 0$. On the contrary, the THz refractive index is commonly equal to or larger than the optical one (i.e., $\Delta k_{THz} \leq 0$). Collinear PM in isotropic crystals [23] and type-0 (eee-type) PM [24] are possible. The corresponding medium ($n_T \approx n_g$) is called "subluminal" or "weakly superluminal" material [25]. In the general case of $\Delta k_{THz} < 0$, noncollinear geometry [17] could work, even in strong superluminal materials ($n_T$ much larger, e.g., ferroelectric). Second, the THz wavevector is much smaller, and its transverse Fourier component is more prominent. Thus, some configurations such as backward [16] and Cherenkov-type THz emissions [13] can be achieved, which are rarely observed in the optical region. The choice of PM geometry is further enriched, and some of the strict PM conditions are relaxed. In the following sections, different types of PMs are summarized and classified, as seen in Figure 1.

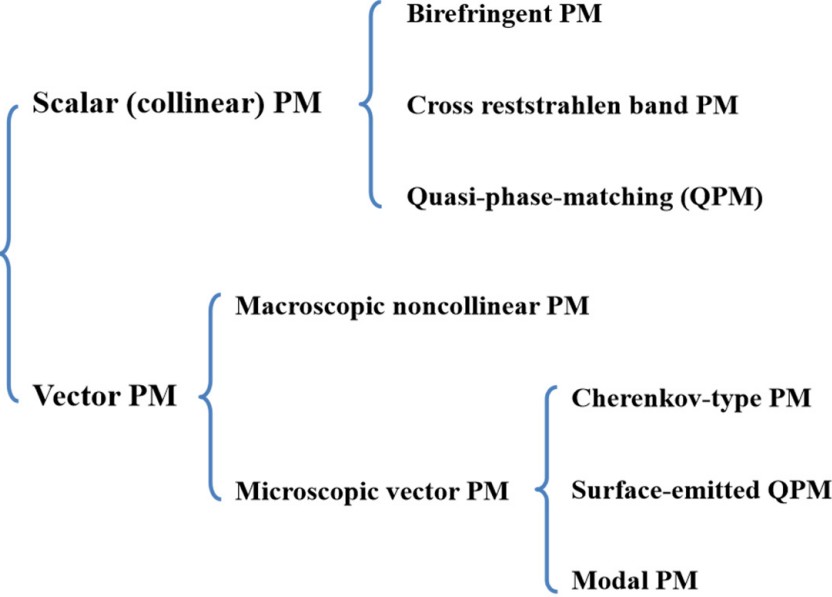

**Figure 1.** Classification of phase-matching in nonlinear THz-wave generation.

## 3. Scalar (Collinear) Phase-Matching

A collinear interaction is favorable due to the overlapping area of the large beams and the ease of alignment. The inherent difference between the refractive indices can be com-

pensated for by the birefringence or cross-reststrahlen band in bulk crystals or by artificial periodic poling/inversion structures. In major situations [15,23,24,26], THz outputs are in the same direction as the input beams (forward coupling); in minor situations [16,27], outputs are in the opposite direction (backward coupling).

Birefringent Phase-Matching

Birefringent PM has been widely adopted in nonlinear processes in optical regions. One of its merits is its broadband and frequency-agile tunability that can be achieved by rotating the crystal. In some traditional non-oxide infrared crystals (such as GaSe [15,16,28–34] and ZnGeP$_2$ [35–38]), birefringent PMs are extended to THz DFG.

In the past 20 years, GaSe (negative uniaxial) crystals have attracted a large amount of attention in nonlinear THz-wave generation due to their extremely low THz absorption and collinear PM. Two forward configurations have been investigated for different frequency band generation techniques [28]. In the scheme for low-frequency bands (Figure 2a), the input pump (1.06 μm) and seed (1.06–1.09 μm) lights are ordinary (slow) and extraordinary (fast), delivering a fast THz-wave (0.0848–5.15 THz). The band is labeled as o-e → e or oee-type, with an effective nonlinear coefficient of $d_{22}\cos^2\theta\cos3\varphi$ and an optimal $\varphi$ of 0°. In fact, the element $d_{16}$ works in the nonlinear coupling and is equal to $-d_{22}$ according to the crystal symmetry. The scheme for the high-frequency bands (also regarded as mid-infrared) is shown in Figure 2b, and the bands are labeled as eoo-type. A tuning range of 7.81–111 THz (38.4 to 2.7 μm) can be obtained by rotating the crystal in the range from 35° to 75°. Backward THz-wave DFG has also been demonstrated with GaSe (Figure 2c) [16]. An external PM angle $\theta_{ex}$ from 15° to 61° corresponds to a tuning range of 0.146–1.79 THz.

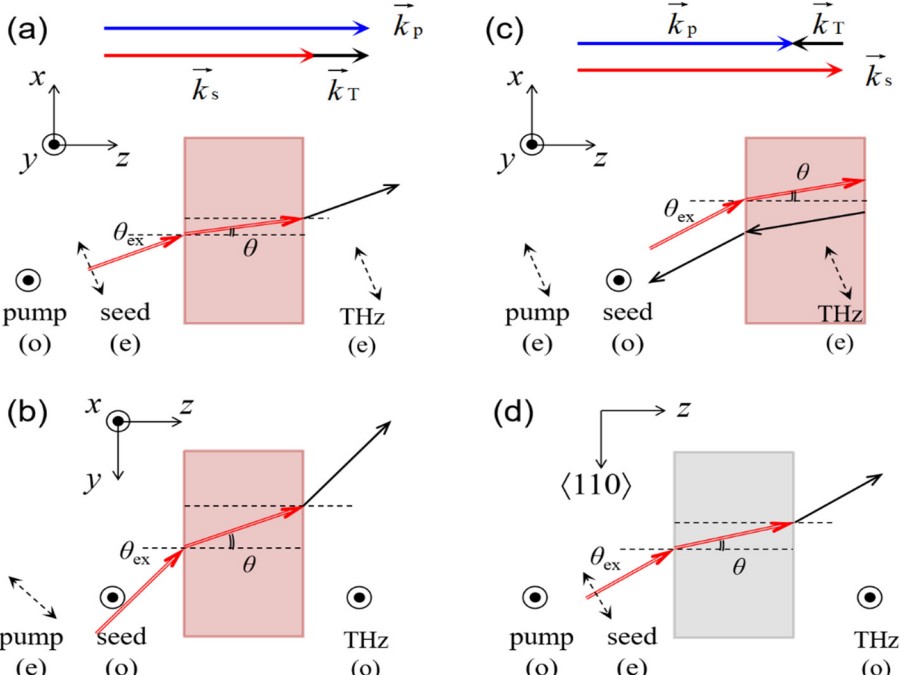

**Figure 2.** Schematic diagram of birefringent PM configurations: (**a**) oee-type, (**b**) eoo-type, (**c**) backward in GaSe, and (**d**) oeo-type in ZnGeP$_2$.

In a positive uniaxial ZnGeP$_2$ crystal [36], 1.06 μm pumped oee-type (similar to Figure 2a) and oeo-type (Figure 2d) PMs provide tuning ranges of 0.18–3.61 and 0.21–3.74 THz with $d_{eff} = d_{36}\sin2\theta\cos2\varphi$ and $d_{36}\sin\theta\sin2\varphi$, respectively. It should be noted that the o- and e-lights in ZnGeP$_2$ are fast and slow lights, respectively. Moreover, birefringent PMs via other d-tensor elements such as the $d_{12}$ of DAST [39] and the $d_{15}$ and $d_{22}$ of LiNbO$_3$ (both forward and backward) [40,41] have been reported.

Angle tuning has been used in most of the studies above. As a critical PM with small angular acceptance [34], the precise control and simultaneous adjustment of the pump wavelength and PM angle are necessary. Temperature tuning has rarely been utilized in noncritical PMs because THz absorption greatly increases with temperature [42,43]. An advantage of birefringent PMs is that there are multiple choices of pump sources. For example, GaSe-DFGs have been pumped at wavelengths of ~1 μm [15,32], ~1.5 μm [31], and ~2 μm [34]. A drawback is the limited kinds of media. The mechanical fabrication and growth of high-quality GaSe are still difficult. Recently developed ε-GaSe [44] and S-doped GaSe [45] have improved the optical properties.

Cross-Reststrahlen Band Phase-Matching

This type of PM is named after the fact that THz and pump frequencies lie across the absorption band [46]. In some materials with proper phonon–polariton dispersion, the mismatch of the refractive indices can be minimized without birefringence. It has been used in isotropic zinc blende [23] and in organic crystals (also called type-0/eee-type PM) [24].

Zinc blende crystals are an excellent THz medium owing to their high nonlinearity, acceptable THz absorption, and well-developed growth and fabrication technique. This type of PM overcomes the drawback of these crystals: their intrinsic isotropy, if the pump wavelength is properly chosen. For GaP (with a transverse optical phonon frequency at 10.9 THz), the optimally matched wavelength lies at 0.9–1 μm and is delivered from an optical parametric oscillator (OPO) [23]. Commercial 1.06 μm laser-pumped GaP collinear DFG has been presented in a number of reports [47–49]. Although it is not perfectly matched, the coherence length ($l_c = \pi / |\Delta k|$) is sufficiently large, and the pump source is much more powerful. Alternative materials such as GaAs and ZnTe have transverse optical phonon frequencies at 8.1 THz and 5.32 THz and an optimally matched wavelength at ~1.3 μm and ~0.8 μm [50]. These crystals should be cut by $\langle 110 \rangle$ or $\langle 111 \rangle$ to utilize the element $d_{14}$. A comprehensive comparison between several nonlinear crystals in collinear DFG was carried out in [51].

Organic crystals with a "D-π-A" molecule/ion structure have been invented and have been applied as THz emitters [52], featuring extremely large proportions of nonlinear diagonal elements $d_{11}/d_{33}$ and a favorable eee-type PM. A main discrepancy of these inorganic crystals is their multiple absorption peaks. In some red-colored organic crystals (DAST [24], DSTMS [53], and OH1 [54]), the matched pump wavelength lies in the range of 1.2–1.6 μm; in yellow-colored crystals (BNA [55]), the matched pump wavelength is in the range of 0.8–0.9 μm. Apart from the OPOs [24,53–57], different kinds of lasers have been employed in this band, including Nd-doped solid-state lasers [58–60], Er-doped fiber lasers [61], and Cr:Mg$_2$SiO$_4$ lasers [62]. The high nonlinearity and THz absorption can be attributed to a short optimal interaction length (crystal thickness commonly below 1 mm [53,63]). As a result, the extremely wide tuning range (up to 30 THz) spans over the absorption bands (those of inorganic crystals are generally limited to below the transverse optical phonon frequency). Moreover, the dependence on the pump wavelength becomes insensitive [53,54], and the synchronized and precise control during frequency tuning is not necessary. The cascaded DFG process is possible when using this type of PM [63,64], which can overcome the quantum defect-related limitations of THz-DFG. An additional potential benefit of organic crystals lies in their designable molecule/ion structure, which allows us to synthesize new crystals with better optical and growing properties (e.g., the derivatives DSTMS, DASB [65], and DASC [66] as well as hydrogen-bonded MLS [67]).

Quasi-Phase-Matching (QPM)

A periodical structure with alternating sign of $\chi^{(2)}$ can extend the effective conversion length, even if the inherent coherence length is small. This configuration is called QPM and is a feasible solution to the PM problem in three kinds of media: isotropic, strongly superluminal, and those with $\Delta k > 0$. The structure can be fabricated by periodically poling a bulk crystal (ferroelectric PPLN [26,27]) or by stacking inverted layers (such as

GaP [11,68], GaAs [69,70], and OH1 [71]). The poling/inverting period $\Lambda$ should be twice the coherence length, and the wavevector mismatch can be compensated for by a grating vector of $2\pi/\Lambda$ (as shown in Figure 3a).

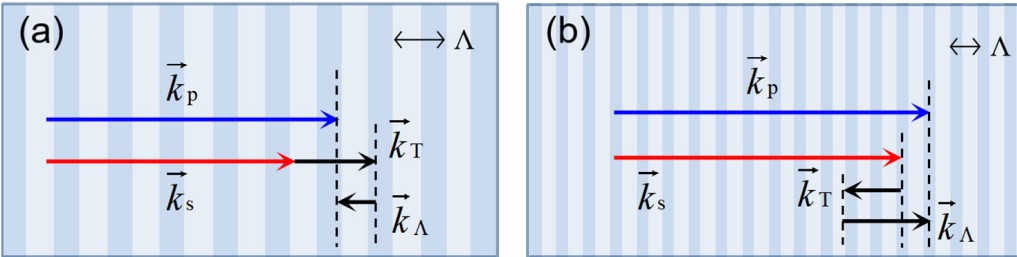

**Figure 3.** Orientation of wavevectors for forward (**a**) and backward (**b**) QPM.

The availability of various $LiNbO_3/LiTaO_3$-based periodical/aperiodical structures, including chessboard-like [72], ridged [73], slant-stripe [74], fanned-out [75], and other pre-designed structures [76,77], have the benefits of ferroelectric properties and the poling technique. The first three geometries will be discussed in Section 4. Other types of QPM crystals based on zinc blende wafers should mainly be fabricated via three methods: optical-contacted, diffusion-bonded, and orientation-patterned. A given period $\Lambda$ commonly matches with one specific frequency and leads to a narrow tuning range [78]. Fanned-out structures [75] and high-order QPM [69] could widen the tuning range to some extent. Some pre-designed configurations with a gradually changing poling period could compress the phase mismatch at different cascading orders caused by crystal dispersion [77]. Continuous-wave (CW) TPO via backward QPM (as seen in Figure 3b) has been reported in PPLN [27].

The optical rectification of ultra-short laser pulses in QPM structures could also give rise to monochromatic THz-waves [78,79]. From the point of view of the time-domain, a multi-cycle wave package is generated by the inverted domains; from the point of view of the frequency-domain, one single frequency is selected by the PM condition. Another DAST-$SiO_2$ multi-layer structure (slightly different from QPM) has been proposed to enhance the single-cycle THz-wave output via phase correction [80].

## 4. Vector Phase-Matching

All of the configurations in this category utilize vector triangles and can help to solve scalar PM problems when $\Delta k < 0$. In other words, vector PM works if and only if the optical group index is smaller than the THz index. As a matter of fact, this condition is valid in a large number of materials. A variety of vector PM geometries have been invented [13,14,17,19]. We can classify them into two divisions: macroscopic noncollinear PM and microscopic vector PM. In the former case, two optical beams (commonly collimated) are obviously noncollinear and deliver a THz beam in a third direction (as seen in [17] and in the *k*-vector diagram in Figure 4a). In the latter case, optical beams (sometimes tightly focused) propagate in approximately the same direction, and vector PM occurs between the spatial Fourier components (such as the Cherenkov-type components [13]). Some of these methods were initially adopted for optical rectification and were later applied to dual-wavelength DFG. Here, "macroscopic" and "microscopic" are used to distinguish these divisions due to the peculiarity of THz-waves. In the traditional optical region, only the former exists, and the "vector" PM is equivalent to the "noncollinear" PM. The prominent transverse THz wavevector components contribute to the microscopic vector PM.

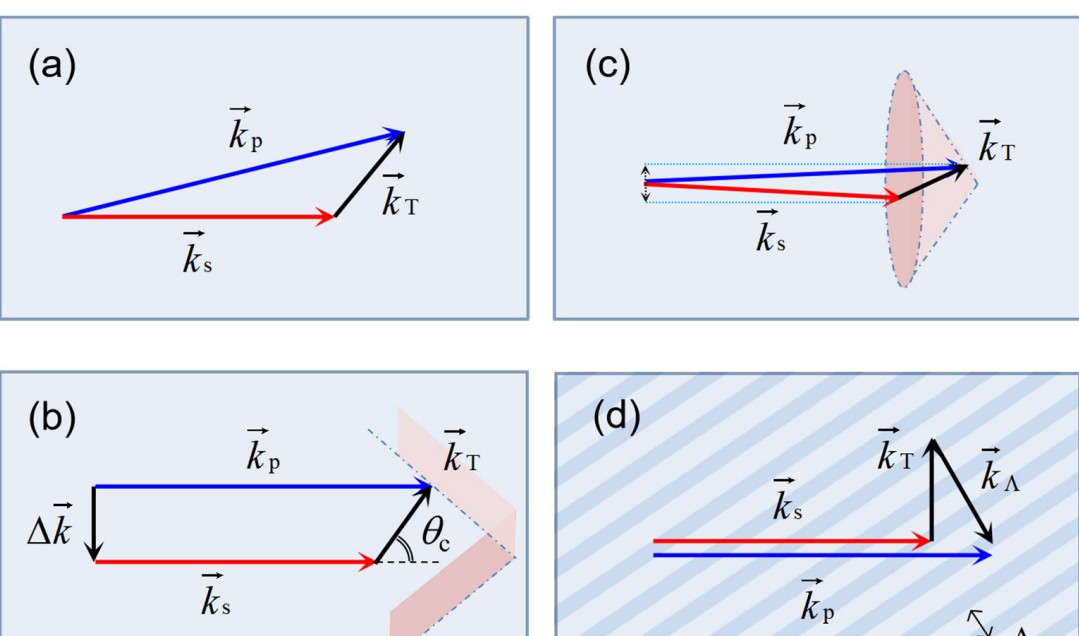

**Figure 4.** *k*-vector diagrams of different types of vector PM: (**a**) macroscopic noncollinear PM, (**b**) Cherenkov-type PM, (**c**) PM pumped by Bessel beam, and (**d**) surface-emitted QPM with slant-stripe-type poling period.

*4.1. Macroscopic Noncollinear Phase-Matching*

Noncollinear laser-pumped THz-wave generation began being investigated as early as 1969 [81]. This configuration is suitable for ferroelectrics, zinc blende, and $KTiOPO_4$ (KTP)-type media. A small angle between the two optical beams (generally below 5°) is sufficient to eliminate the wavevector mismatch, which should be synchronously changed during frequency tuning. To efficiently couple out the THz beam, traditional rectangular crystals are usually cut at an angle [82] or are attached with a Si prism [17].

As discussed above, the pump wavelength of isotropic crystals in collinear PM is determined by the dispersion property, which greatly limits the choice of pump source. This noncollinear approach makes it possible to match with a wide range of optical wavelengths. For example, 2.97 THz [82], 0.5–3 THz [83], and 0.11–4.15 THz [84] were generated using GaAs pumped by $CO_2$ lasers. GaP can provide a wider tuning range (up to 6 THz) due to its higher transverse optical phonon frequency. Experiments have been performed with pump sources such as Nd:YAG lasers and OPO [85], $Cr:Mg_2SiO_4$ lasers [86], and fiber amplifiers [87–89]. Compared to the ~0.9 μm OPO [23], some of these lasers have better performance, demonstrating higher output power and a narrower linewidth and operation at a higher repetition rate or CW. A frequency-domain THz spectrometer based on CW GaP-DFG equipped with power/frequency feedback, motorized stages, and a Si bolometer with a high spectral resolution was constructed and applied for the quality evaluation of pharmaceuticals [8].

$LiNbO_3$ crystals are the most studied crystal type in TPO [10,17,90–93] and is-TPG [9,94–98], as they can take advantage of the $A_1$-symmetry polariton mode. TPO only requires a single- and fixed-wavelength input; commercial and compact Q-switched Nd:YAG lasers are available. The THz frequency is tuned continuously by rotating the cavity of the Stokes light. Progress has been made in surface-emitted geometry via internal total reflection [10,92], pump recycling [90], and the ring cavity [91]. CW-TPO was achieved by putting a Si prism-coupled $MgO:LiNbO_3$ crystal within an end-pumped $Nd:GdVO_4$ laser resonator [93]. Later, sub-nanosecond (sub-ns) or picosecond (ps) lasers were employed as the pump source, combing a tunable seed and forming an is-TPG to pursue high peak power [94–99]. A real-time spectroscopic measurement system based on a multi-frequency

is-TPG was set up by inputting five seed beams simultaneously [9]. Noncollinear DFGs have also been conducted in bulk LiNbO$_3$ crystals using, for example, a CW fiber-laser pump [100] and a reflected signal beam [101].

A similar TPO configuration was also extended to Raman-active KTP-type crystals (such as KTiOPO$_4$ [102], KTiOAsO$_4$ [103], and RbTiOPO$_4$ [104]). The high optical damage threshold allows us to use high pump power and to obtain high nonlinear gains. Due to the infrared absorption by the A$_1$ crystalline modes in these crystals, discrete tuning bands were observed. A green laser pump moved the tuning bands toward higher frequencies [105].

### 4.2. Microscopic Vector Phase-Matching

#### Cherenkov-Type Phase-Matching

Classical Cherenkov radiation with a conic wavefront was first discovered in an experiment on the glow of uranyl salt solutions under gamma-ray irradiation in 1934 [106]. A similar radiation form excited by an electromagnetic pulse (called "optical Cherenkov radiation") was predicted in 1962 [107] and was experimentally observed in optical rectification in 1984 [108]. Later, it was regarded as a kind of PM [109]. Since 2008 [13], Cherenkov-type PM in monochromatic THz-DFG has been studied. Different from the classical/optical Cherenkov radiation explained by Huygens' principle, this Cherenkov-type PM is essentially a type of vector PM (shown in Figure 4b). It has been considered that PM is automatically satisfied under a certain angle with respect to the laser path ($\cos\theta_c = n_g/n_T$). General pump beams with a finite size consist of a continuum of radial wavenumbers that create the transverse components of Cherenkov wavevectors as long as the beam is focused into a specific size according to the THz wavelength.

This kind of PM has been demonstrated in ferroelectric LiNbO$_3$ [13,110–113] and in organic DAST [114] crystals. Theoretical models based on the analytical solution via spatial Fourier expansion [18] and coupled wave equations via the split-step method [115] have been reported. Line-focused pump beams can generate a wedged THz wavefront [116], which is better to collect than a conic one. Similar to in Section 4.1, Si prism coupling can decrease the absorption and increase the output, especially at high frequencies [117]. Moreover, real-time sensing with a Cherenkov-type evanescent wave has been achieved [118]. This "automatical" PM has a large tolerance (free from the precise control of input parameters) and is almost independent of the pump wavelength [119]. Strictly speaking, it is not a perfect PM type because of the transverse mismatch [110]. A small pump beam width corresponds to a wide tuning range and high efficiency. This means that a good pump beam quality is favorable. On the other hand, a narrow focus causes severe divergence, decreasing the effective radiation length. This contradiction can be solved by employing a leaky-mode waveguide, which will be discussed in the following sections.

Another PM geometry with Bessel-type pump beams was introduced [19]. By properly controlling the input beam profile (as well as the longitudinal wavenumber), oblique THz wavevectors can be generated via vector PM (as seen in Figure 4c), forming a conic wavefront. Compared to the Cherenkov-type PM above, the radial Fourier component of the Bessel pump beam is a single value that should strictly obey the PM conditions in Figure 4c. Using pump beams with a tilted phase front is an alternative approach for PM in superluminal materials [14] and is extended from optical rectification [120].

#### Surface-Emitted QPM with 2D Poling Period

In Section 3, collinear QPM with a longitudinal poling period was presented. A well-developed PPLN crystal has been widely used in TPO and DFG. One of the drawbacks is the high THz absorption, which limits the effective length. Surface emission achieved by tightly focusing the pump beam was proposed to minimize absorption [121,122]. The perpendicular THz wavevector originates from the transverse Fourier component (similar to Cherenkov-type PM). It still suffers from the contradiction between transverse mismatch and beam divergence. A two-dimensional (2D) poling period provided a transverse grating vector (e.g., Figure 4d), and so tight focusing was no longer necessary. Chessboard-like [72]

and ridged [73] PPLN-based DFG have been demonstrated experimentally. Another domain structure, walling along the propagation direction of optical beams, was designed, which delivered a THz beam at around the Cherenkov angle [123]. Cavity-enhanced CW THz DFG was achieved with a slant-stripe-type PPLN pumped by vertical external cavity surface-emitting lasers [74]. Similar poling structure also benefited a novel cavity-less backward TPO [124].

Modal Phase-Matching in a Waveguide

A nonlinear waveguide is a substitute for bulk crystal-based THz emitters due to modal PM and optical field confinement. By introducing the size parameter (transverse mode distribution), the THz effective index can be changed at will, to match the optical waves. As early as 1974 [12], a planar dielectric GaAs waveguide pumped by $CO_2$ lasers had been reported. Components made of zinc blende materials, including planar [125,126], rod [127], and ridge-type [128] components, commonly operate in guided-mode. THz output is coupled in the forward direction, and it is sometimes treated as "collinear". However, it is intrinsically a kind of vector PM (between longitudinal propagation constants). A comprehensive comparison between different zinc blende materials for dielectric waveguide creation has been presented [129], deriving a general figure of merit for the evaluation of conversion efficiency. All-fiber CW THz generation was demonstrated experimentally based on thermal poling-induced second-order nonlinearity [130]. In these guided-mode PMs, the tuning ranges are relatively narrow, as determined by the distribution of the intrinsic mode. Another drawback lies in the coupling efficiency of the pump into the nonlinear core (small aperture), which limits the total output power.

Leaky-mode (sandwich-like) waveguides have also been applied for nonlinear THz-wave generation, which is the combination of Cherenkov-type PM and the waveguide structure [131–134]. The optical fields are confined in a high-nonlinear core (e.g., $MgO:LiNbO_3$) with a small cross section and deliver a leaky-mode THz-wave coupled into a Si clad. The tuning range is wider and flatter [132] due to the continuously distributed leaky mode. A room-temperature CW THz spectrometer was constructed based on this configuration, which offered broad spectral coverage (up to 7.5 THz), a high spectral resolution, a μW-level emission power, and an absolute frequency reference [135]. Intracavity THz DFG in a dual-wavelength mid-infrared quantum cascade laser (QCL) was achieved [136]. The QCL was designed to have giant optical nonlinearity in the active region and to form a multi-layer leaky-mode waveguide, which offered efficient nonlinear conversion and output coupling.

Other Phase-Matching Configurations

There are some PM geometries that are not included in the categories above. The concept of phase correction, a situation in which the velocities of the interactive waves are not perfectly matched, was introduced in 1962 [137]. A phase shift of 180° could help to circumvent the mismatch, which is possible at the total reflection point [138] or within a Fabry–Perot interferometer (called "cavity PM") [139,140]. Triply resonant DFG has also been analyzed theoretically [141]. The form of the resonator could be a sheet or a microdisk (in whispering gallery mode) [142]. A hybrid thin-film waveguide-based racetrack resonator was proposed recently, which gave rise to an integrated CW THz source [143].

## 5. Summary and Discussion

As optical THz sources (DFG, TPO, or is-TPG) mainly consist of nonlinear crystals and pump lasers, the output characteristics are greatly determined by these two factors. Different kinds of crystals have been investigated, ranging from traditional infrared ($GaSe/ZnGeP_2$), zinc blende ($GaAs/GaP$), ferroelectrics ($LiNbO_3$), and KTP-type ($KTiOPO_4/KTiOAsO_4$) crystals to organic crystals (DAST/OH1). At present, each kind has its particular advantages and inherent difficulties. An ideal nonlinear medium should exhibit the following merits: high nonlinearity, a wide transparency range, a high optical damage threshold, and fixability for large-size growth and fabrication. Since optical

THz sources are superior due to their continuous and wide tunability, the corresponding pump/seed should generally be tunable. Initially, Q-switched laser-pumped OPOs were employed, which commonly operated in ns-pulses with low repetition rates. Later, tunable lasers with amplifiers at the desired wavelength bands were developed, offering a narrower linewidth, better beam quality, acceptable power, and a high repetition rate or even CW operation.

Phase-matching represents an important connection between the two parts above. A favorable PM configuration could make full use of the merits of current lasers and crystals (some are commercially available and have demonstrated high performance) and could overcome their drawbacks. Traditional infrared crystals are suitable for birefringent PM with angle tuning. Zinc blende crystals could adopt cross-reststrahlen band PM, QPM, nonlinear PM, Bessel-type PM, guided-mode PM, and cavity PM. Ferroelectrics and KTP-type crystals could adopt QPM, nonlinear PM, Cherenkov-type PM, front-tilting PM, and leaky-mode PM. Organic crystals commonly use type-0 PM. As it is highly desired in frequency-domain THz spectroscopy, efficient nonlinear conversion under relatively low pump power is in demand. Integrated devices (e.g., fiber-coupled or on-a-chip) are arising as a new trend. A well-designed PM becomes necessary, which could relax the matching condition in a wide range of crystals and pumps and could finally promote the application of coherent THz sources.

**Author Contributions:** Conceptualization, P.L.; investigation, C.N. and W.L. (Wei Li); resources, W.L. (Weifan Li), Q.F. and L.G.; writing—original draft preparation, P.L. and C.N.; writing—review and editing, P.L., C.N. and Z.L.; visualization, P.L. and C.N.; supervision, F.Q.; project administration, F.Q.; funding acquisition, P.L. and F.Q. All authors have read and agreed to the published version of the manuscript.

**Funding:** This research was funded by the Youth Innovation Promotion Association CAS (2019204); "XingLiaoYingCai" Talents of Liaoning Province (XLYC2007074); and Young and Middle-Aged Scientific and Technological Innovation Talents of Shenyang (RC200512).

**Conflicts of Interest:** The authors declare no conflict of interest.

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
