# Peer review of "Phase-Matching in Nonlinear Crystal-Based Monochromatic Terahertz-Wave Generation"

_crystals, doi:10.3390/cryst12091231_

Round 1
Reviewer 2 Report
The authors report a review paper entitled “Phase matching in nonlinear crystal based monochromatic terahertz-wave generation”.
I think that the manuscript is well established focusing on phase-matching conditions with typical nonlinear crystals for THz-wave generation.
Although the traditional configuration of vector phase-matching is described in section 4, the latest configuration should be also included in the manuscript to promote the improvement of coherent THz sources.
1: Thin film LN is promising for integrated THz photonic circuits. The novel devices would make newly THz applications. "Efficient terahertz generation scheme in a thin-film lithium niobate-silicon hybrid platform," Opt. Express 29, 16477-16486 (2021)
2: Recently, a mirrorless TPO with novel backward noncolinear phase-matching has been reported as "Tunable Backward Terahertz-wave Parametric Oscillation," Sci. Rep. 9, 726 (2019). The feature of “mirrorless” is attractive for scientific as well as industrial applications.
Everyone knows that phase matching is well established to use the great potential of the nonlinear crystal. But it might be still unrevealed as the above comments. I expect that the revised manuscript would encourage researchers in the fields of nonlinear optics as well as optical materials, integrated science, and industrial engineers.
